# A Micropillar Array Based Microfluidic Device for Rare Cell Detection and Single-Cell Proteomics

**DOI:** 10.3390/mps6050080

**Published:** 2023-09-04

**Authors:** Kangfu Chen, Zongjie Wang

**Affiliations:** 1Department of Biomedical Engineering, McCormick School of Engineering, Northwestern University, Evanston, IL 60208, USA; zongjie.wang@northwestern.edu; 2Chan Zuckerberg Biohub Chicago, Chicago, IL 60607, USA

**Keywords:** rare cells, single-cell proteomics, microfluidics, CTC isolation, CTC collection

## Abstract

Advancements in single-cell-related technologies have opened new possibilities for analyzing rare cells, such as circulating tumor cells (CTCs) and rare immune cells. Among these techniques, single-cell proteomics, particularly single-cell mass spectrometric analysis (scMS), has gained significant attention due to its ability to directly measure transcripts without the need for specific reagents. However, the success of single-cell proteomics relies heavily on efficient sample preparation, as protein loss in low-concentration samples can profoundly impact the analysis. To address this challenge, an effective handling system for rare cells is essential for single-cell proteomic analysis. Herein, we propose a microfluidics-based method that offers highly efficient isolation, detection, and collection of rare cells (e.g., CTCs). The detailed fabrication process of the micropillar array-based microfluidic device is presented, along with its application for CTC isolation, identification, and collection for subsequent proteomic analysis.

## 1. Introduction

Over the past decade, the detection and analysis of rare cells have emerged as a crucial and transformative field, holding significant implications for disease diagnosis, treatment, and our understanding of fundamental biological processes [1,2,3,4]. Rare cells are those with low prevalence amidst a large number of background cells, often defined as less than 1000 cells per milliliter sample [5]. Despite their scarcity, these cells play pivotal roles in various biological processes, including immune reactions [6,7,8,9], tissue regeneration [10,11,12], and cancer progression [13,14,15,16]. For instance, circulating tumor cells (CTCs) have proven to be invaluable in providing insights into cancer advancement, metastasis, and treatment responses [17,18,19,20,21,22,23]. In recent years, single-cell related technologies, such as single-cell RNA sequencing and single-nucleus RNA sequencing, have significantly advanced our comprehension of rare cells within complex biological systems [24,25,26,27]. Characterizing single cells has led to the creation of reference cell atlases for entire organisms, and multi-modal measurements at the single-cell level have provided further understanding of cellular processes and states [28,29,30]. The development of single-cell mass spectrometric analysis (scMS) has enabled the study of proteins and post-translational modifications without the need for affinity reagents like antibodies [31]. However, since proteins within cells cannot be amplified, single-cell proteomic analysis is limited by the amount of protein obtainable. Therefore, achieving highly efficient isolation and detection of rare cells becomes crucial as a prerequisite for successful proteomic analysis.

Microfluidics has emerged as a highly effective tool for detecting rare cells, demonstrating remarkable detection efficiency [32,33,34,35,36]. For instance, microfluidic devices equipped with antibody functionalized microchannels can efficiently isolate and capture CTCs from blood samples [37,38,39]. Diverse microfluidic platforms can be employed to detect and analyze rare immune cell populations, such as antigen-specific T cells or B cells, which are critical in the cancer immunotherapy [40,41,42,43,44]. Additionally, microfluidic systems using magnetophoresis have shown superb isolation efficiency of rare cells [45,46,47,48,49,50]. Rare cells can be magnetically labeled by antibody conjugated magnetic nanoparticles (MNPs) and separated from non-target cells when an external magnetic field is applied. While different rare cell capture platforms were developed with high efficiency, the release of captured rare cells is still challenging. To date, different approaches have been proposed for captured CTC release. For example, using aptamer-coated nanotube structure for CTC capture and enzymatic treatment for CTC release [51]; using aptamer-functionalized nanosphere for CTC capture and a thiol exchange reaction for CTC release [52]; using thermal-responsive polymer-grafted nanostructure for CTC capture and temperature shift for CTC release [53]; using biodegradable nano-film for CTC capture and release [54]. However, these surface treatment methods are usually complicated and cannot get full CTC recovery. On the other hand, physical dislodging CTCs can be a simple and efficient alternative. In this work, we propose a micropillar-based microfluidic device designed for highly efficient isolation, detection, and release of circulating tumor cells (CTCs) to enable subsequent proteomic analysis.

## 2. Experimental Design

The proposed micropillar based microfluidic device relies on the specific capture of rare cells such as circulating tumor cells (CTCs) based on their unique surface marker [55]. As illustrated in Figure 1A, antibodies (e.g., anti-EpCAM) are immobilized in the microchannels of the microfluidic device. The whole blood sample is introduced to the microfluidic device. CTCs expressing EpCAM will be specifically captured in the microchannel. The micropillar array patterned in the microchannel increases the interaction between CTCs and antibodies, making them more likely to be captured. Also, the micropillars are localized within the pillar and can be used for the localization of captured CTCs. After sample processing, captured CTCs will be detected within the microfluidic device using a fluorescence microscope. The identified CTCs will be marked and released using a PDMS punch. The collected CTCs will be used for subsequent single-cell proteomic analysis (Figure 1B).

### 2.1. Materials

AZ 400k developer (MicroChemals GmbH, Ulm, Germany)Chromium etchant (TechniEtch Cr01, MicroChemals GmbH, Ulm, Germany)5-inch chromium coated glass plate covered with positive photoresist film (Photomask Portal, Richardson, TX, USA)4-inch silicon wafers (University Wafer, South Boston, MA, USA)Tweezers for 4-inch silicon wafer (Catalog No. 18-100-946, Fisher Scientific, Newington, NH, USA)Acetone (Sigma Aldrich, St Louis, MO, USA)Isopropanol (IPA) (Sigma Aldrich, MO, USA)6-inch wide-open round beakers (Fisher Scientific, NH, USA)N2 dryerHexamethyldisiloxane (HMDS) (SKU 440191, Sigma Aldrich, MO, USA)SU8 3035 photoresist (Kayaku Advanced Materials, Westborough, MA, USA)SU8 developer (Kayaku Advanced Materials, MA, USA)Polydimethylsiloxane (PDMS) Sylgard 184 kit: silicone elastomer base and curing agent (Dow Corning, Midland, MI, USA)Trichloro(1H,1H,2H,2H-perfluorooctyl)silane (TPOS) (SKU 448931-10G, Sigma Aldrich, MO, USA)Weighing dish (Catalog No. 01-549-750, Fisher Scientific, NH, USA)Petri dish (15 cm) (Catalog No. 09-720-500, Fisher Scientific, NH, USA)Weighing scale (Catalog No. 01-922-341, Fisher Scientific, NH, USA)Razor blade or scalpel (Catalog No. NC0134996, Fisher Scientific, NH, USA)PDMS punches (Catalog No. 12-460-402, 12-460-410, Fisher Scientific, NH, USA)Microscope slides (Catalog No. 12-550-102, Fisher Scientific, NH, USA)99% ethanol (SKU 459836, Sigma Aldrich, MO, USA)Deionized (DI) water1 mL pipet tips (Catalog No. 05412307, Fisher Scientific, NH, USA)Stainless steel dispensing needle with luer lock connection (Catalog No. 11-101-1671, Fisher Scientific, NH, USA)Cleaned and bagged high-purity white silicone rubber tubing (Catalog No. 51845K51, McMaster Carr, Elmhurst, IL, USA)PTFE light wall tubing (Part No. STT-21, Component Supply Co., Sparta, TN, USA)3D printed supporting structureSterile Luer-Lock syringes (3 mL and 5 mL) (Fisher Scientific, NH, USA)Phosphate-buffered saline (PBS) without Ca^2+^/Mg^2+^ ions (311-010-CL, Wisent Bioproducts, Saint-Jean-Baptiste, QC, Canada)Avidin (Catalog No. MP215004705, Fisher Scientific, NH, USA)Bovine serum albumin (BSA) (SKU A7030, Sigma Aldrich, MO, USA)Biotinylated anti-EpCAM (eBioscience, San Diego, CA, USA)Cell line (e.g., PC9 cells, ATCC)Cell culture medium (e.g., RPMI1640 medium,)Fetal bovine serum (FBS) (080-150, Wisent Bioproducts, QC, Canada)Trypsin-EDTA (0.25% Trypsin with EDTA 4Na) (325-042-CL, Wisent Bioproducts, QC, Canada)Vybrant dye DiD (Catalog No. V22887, Thermofisher, Waltham, MA, USA)Sterile 1.7-mL centrifuge tubes (Catalog No. 02-682-000, Fisher Scientific, NH, USA)

### 2.2. Equipment

Maskless Aligner (Heidelberg uPG 501, Heidelberg Instruments Mikrotechnik GmbH, Heidelberg, Germany)Spin coater (H6-23, Laurell Technologies, Lansdale, PA, USA)Digital Programmable Stirring Hot Plates (HS40/HS40A, Torrey Pine Scientific, Carlsbad, CA, USA)Contact Mask Aligner (Suss MA6 Mask Aligner)Desiccator with a vacuum connection (Catalog No. 08-594-15A, Fisher Scientific, NH, USA)Programmable Oven (SKU 2958M5, Medicus Health, Kentwood, MI, USA)Plasma treater (SKU 12051A, Electro-Technic Products, Chicago, IL, USA)Programmable, dual-channel infusion and withdrawal modular syringe pump (Fusion 200-X, Chemyx, Stafford, TX, USA).Pipet (P10, P200, P1000) (Eppendorf) (Fisher Scientific, NH, USA)Serological pipet (Catalog No. 9541, Thermofisher, USA)CO_2_ incubator for cell culture (Catalog No. 11-676-604, Fisher Scientific, NH, USA)Biosafety hood (1300 Series, Fisher Scientific, NH, USA)Centrifuge (Eppendorf) (Catalog No. 05-413-112, Fisher Scientific, NH, USA)Mini centrifuge (Catalog No. 75002541, Fisher Scientific, NH, USA)

## 3. Procedure

This section includes three main parts: (1) the microfabrication of the micropillar based device; (2) the functionalization of the device using antibodies; (3) the process of samples using the microfluidic device and the release of captured CTCs.

**CRITICAL STEP** The fabrication of the micropillar based device is critical as its performance largely depends on the fabrication outcome. As shown in Figure 2, the fabrication includes three major steps. First, a silicon mold containing the micropillar array feature is fabricated using photolithography (Figure 2A). Next, the PDMS substrate and the PDMS thin-film cover are fabricated using soft lithography (Figure 2B). Third, the PDMS substrate and PDMS thin-film are plasma-treated and permanently bonded (Figure 2C).

### 3.1. Chrome Mask Writing

Draw the layout of the micropillar-based microfluidic device using CAD software and convert the file into DXF form.Load the chromium and positive photoresist coated glass plate to the maskless aligner.Pattern writing on the photoresist layer of the glass plate using the maskless aligner.Dilute the AZ 400K developer 5 times (e.g., 40 mL of AZ 400K developer and 160 mL of DI water).Develop the glass plate with the diluted AZ 400K developer for 20 s.Rinse the glass plate with running DI water for 10 s.Soak the glass plate in the chrome etchant for 1 min.Rinse the glass plate with running DI water for 10 s.Strip off the undeveloped photoresist in undiluted AZ 400K developer for 30 s.Rinse the chrome mask with running DI water for 10 s.Dry the chrome mask with the N_2_ gun.

### 3.2. Silicon Mold Fabrication

12.Bath a 4-inch silicon wafer in acetone for 10 min.13.Spray wash the silicon wafer with IPA to remove acetone residues.14.Bake the wafer at 120 °C for 10 min until fully dry and cool it down to room temperature.15.Vaporize HMDS onto the silicon wafer to create hydrophobic surfaces.16.Spin-coat a layer of 50 µm-thick SU-8 3035 photoresist (2000 rpm for 30 s).17.Soft bake the wafer on a hotplate at 60 °C for 1 min.18.Ramp up the hotplate temperature to 95° C at 450 °C/h.19.Incubate the wafer at 95 °C for 10 min.20.Expose the wafer using the contact mask aligner (200 mJ/cm^2^).21.Post exposure bake the silicon wafer at 95 °C for 5 min.22.Develop the silicon wafer in SU-8 developer for 7 min.23.Hard bake the wafer at 125 °C in the oven for 15 min.

### 3.3. Micropillar-Based Microfluidic Device Fabrication

24.Place the silicon mold with a small weighing bowl in the vacuum desiccator.25.Dispense 20 µL of TPOS in the weighing bowl.26.Turn on the vacuum for 1 min to create a low-pressure environment for TPOS evaporation.27.Incubate the silicon mold for 30 min.28.Place the silicon mold in a 150 cm petri dish.29.Mix 30 g of PDMS prepolymers (base: curing agent = 10:1) in a weighing boat.30.Place the PDMS mixture in a vacuum for 15–30 min to remove bubbles.31.Dispense PDMS mixture on the silicon mold.32.Incubate PDMS in the oven at 75 °C for 2–4 h.33.Cut off the polymerized PDMS substrate with fluidic feature from the silicon mold using a scalpel or razor blade.34.Punch holes at the inlet and outlet of the PDMS substrate using a punch with inner diameter of 1.2 mm.35.Treat an unused silicon wafer with TPOS as described from Step 24–27.36.Apply 10 g of degassed liquid PDMS mixture on the silicon wafer.37.Spin the wafer at 1000 rpm for 1 min.38.Incubate the PDMS coated silicon wafer at 100 °C for 5 min.39.Treat the PDMS substrate with an oxygen plasma treater for 1 min.40.Treat the PDMS film on the silicon wafer with the oxygen plasma treater for 1 min.41.Bind the PDMS substrate with the PDMS film.42.Incubate the sandwiched layers at 100 °C for 1 min in the oven.43.Cool the bonded device down to room temperature.44.Gently cut off the excessive PDMS thin film using the scalpel.45.Peel the microfluidic device off the silicon wafer.46.Physically attach the microfluidic device on a microscope slide for sample processing.

### 3.4. Antibody Immobilization in the Microfludic Device

47.Prepare 2 sections of silicone rubber tubes using a scalpel or razor blade. One with a length of 5 cm. The other with a length of 10 cm.48.Prepare 2 sections of PTFE light wall tubes with a length of 8 mm using a scalpel or razor blade.49.Couple the silicone rubber tubes with the PTFE light wall tubes by inserting the PTFE light wall tube to one end of the silicone rubber with a depth of 4 mm.50.Connect the short-length silicone rubber tube to the inlet of the chip by inserting the coupled PTFE light wall tube.51.Connect the long-length silicone rubber to the outlet of the chip by inserting the coupled PTFE light wall tube.52.Connect the short-length silicone rubber tube to a 1 mL pipet tip used as a sample reservoir. Keep the pipet tube vertical to the chip using a 3D printed hanger or fixture.53.Fix a 5 mL Luer-locked syringe on the syringe pump.54.Cap the syringe with a blunt-tip mixer nozzle.55.Connect the long-length silicone rubber tube to the blunt-tip mixer nozzle.56.Introduce 300 µL of 99% ethanol to the sample reservoir.57.Withdraw 250 µL the ethanol at a flow rate of 2 µL/s using the syringe pump.58.Introduce 200 µL of DI water to the sample reservoir and withdraw 200 µL at a flow rate of 2 µL/s.59.Introduce 200 µL of PBS to the sample reservoir and withdraw 200 µL at a flow rate of 2 µL/s.60.Introduce 100 µL of avidin to the sample reservoir and withdraw 100 µL at a flow rate of 2 µL/s.61.Incubate for 15 min at room temperature.62.Introduce 300 µL of PBS to the sample reservoir and withdraw 300 µL at a flow rate of 2 µL/s.63.Dilute anti-EpCAM 25 times (4 µL of antibodies in 100 µL of PBS at a concentration of 20 µg/mL)64.Introduce 100 µL of anti-EpCAM to the sample reservoir and withdraw 100 µL at a flow rate of 2 µL/s.65.Incubate for 20 min at room temperature.66.Introduce 300 µL of 1% BSA to the sample reservoir and withdraw 300 µL at a flow rate of 2 µL/s.

### 3.5. Capture of Rare Cells from a Spiked Sample

67.Trypsinize PC9 cancer cells by adding 5 mL of 0.25% trypsin-EDTA followed by incubating for 5 min at 37 °C.68.Neutralize the trypsin-EDTA by adding 10 mL of whole culture medium (RPMI 1640 supplemented with 10% FBS and 1% penicillin streptomycin).69.Spin down the cells at 1200× *g* for 4 min.70.Discard the supernatant and wash the cells with 10 mL of PBS at 1200× *g* for 4 min.71.Discard the supernatant and resuspend the cells in 1 mL of PBS.72.Determine the concentration of cells using a hemocytometer.73.Aliquot 1 mL of PC9 cells in PBS at a concentration of 1 × 10^6^ cells/mL.74.Add 5 µL of Vybrant DID dye in the cell sample and mix cell by pipetting.75.Incubate the mixture at 37 °C for 15 min.76.Spin down the cells at 1200× *g* for 4 min.77.Wash the cells 3 times with 1 mL of PBS.78.Resuspend the cells in 1 mL of PBS.79.Determine the concentration of cells using a hemocytometer.80.Aliquot 1 mL of cell sample with a concentration of 10^3^ cells/mL through serial dilution.81.Add 50 µL of cells into 1 mL of whole blood sample supplemented with 1.8 mg/mL EDTA and mix well.82.Load the spiked sample to the sample reservoir of the microfluidic device.83.Withdraw 1.05 mL of the sample at a flow rate of 1 µL/s.84.Wash the microfluidic device by withdrawing 300 µL of PBS at a flow rate of 2 µL/s.85.Repeat Step 83 two times.86.Clamp the silicone tubes of both inlet and outlet.87.Dislodge the microfluidic device from the sample reservoir and the syringe.88.Inspect the microfluidic device under a fluorescence microscope. Fluorescently labeled PC9 cell can be detected under the APC channel.89.Mark the positions with captured PC9 cells using a marker.90.Detach the microfluidic device from the microscope slide.91.Punch though the microfluidic device of the captured PC9 position using a PDMS punch with an inner diameter of 5 mm.92.Collect the dislodged piece of the PDMS with captured PC9 cells in a 1.7 mL centrifuge tube.93.Fill the centrifuge tube with 1 mL of cell culture medium to keep the viability of the captured PC9 cells

## 4. Expected Results

This work is to develop a microfluidic system for the isolation of rare cells such as CTCs for single cell proteomic analysis. The success of the system relies on three aspects: the fabrication of the microfluidic device, the functionalization of the microfluidic device and the detection of rare cells. For the fabrication of microfluidic device, the essential component is the silicon mold. To ensure the quality of the silicon mold, the surface of the silicon wafer should be cleaned thoroughly. The acetone can efficiently remove organic contamination from the silicon wafer. In some occasions (e.g., SU8 residue contamination), the silicon wafer can be cleaned with piranha solution. However, piranha solution is highly corrosive, users should be properly trained and handle the solution with extra caution. To strengthen the adhesion of SU8, HMDS vapor is used to create a hydrophobic surface on the silicon wafer. Alternatively, the silicon wafer can be treated with buffered oxide etch (BOE) to remove silicon oxide and a superior hydrophobic surface can be made. The channel height of the microfluidic device is 50 µm which requires a relatively thick SU8 layer. Therefore, the SU8 coated silicon wafer should be handled with patience. To prevent bubbles, the SU8 3050 bottle can be heated in water bath at 60 °C for 10 min. To achieve optimal uniformity, the SU8 coated silicon wafer can be baked at 60 °C for 10 min and ramp the baking temperature up to 95 °C at a ramping rate of 120 °C/h. For exposure, following the manufacturer’s protocol is optimal. It is advised not to over exposure SU8 which will make the photoresist more fragile and less adhesive to the silicon wafer. The development of unexposed SU8 is critical. Over development can cause feature damage and detachment, while not sufficient development can leave significant residue on the silicon surface which will affect the later process. Hard bake after development is preferred. The thermos cracks during SU8 development will disappear after the hard bake. The success of the silicon mold can be determined by inspection using an optical profilometer. The measurement of the channel height can be performed. The good mold should have a relatively uniform channel height in different positions of the silicon mold and the variation should be smaller than 5 µm.

For the functionalization of the microfluidic device, preventing bubbles is important. When withdrawing ethanol, tapping the microfluidic device to drive the bubbles out can be a good trick for bubble removal. Ethanol is used for degassing the microfluidic device, it should be washed out thoroughly with DI water. Redundant ethanol can cause the precipitation of salt in PBS. The functionalization of the microfluidic device with antibodies allows the microfluidic device to capture rare cells by targeting their specific markers. The antibody immobilization can be inspected by introducing excessive amount of fluorescently labeled secondary antibodies. A fluorescence signal calibration curve can be generated to determine the saturated concentration of primary antibodies.

When processing blood samples, it is advisable to gently agitate the sample in the inlet reservoir with a pipette to regularly to keep the homogeneity of the sample and prevent clogging near the inlet of the device. The detection of rare cells is through the identification of specific biomarkers on the rare cells. The protocol here shows the process of a spiked sample which simulate actual clinical samples, and the target cells are pre-labeled. However, for actual clinical sample processing, labeling of rare cells are usually performed after sample processing. For example, some CTCs can be labeled with fluorescently tagged anti-cytokeratin (CK) and DAPI, while normal white blood cells can be labeled with fluorescently tagged anti-CD45 and DAPI. In this case, CTCs can be detected as CK+/DAPI+/CD45−. When a target cell is detected and located, its position can be roughly marked. If the device works properly, the capture efficiency should be more than 90% and the purity should be higher than 70% [37,56,57]. It is expected to detect less than 10 target cells in 1 mL of whole blood [37]. The performance of the device can be affected by different factors. First of all, the blood sample should be fresh or properly prepared with EDTA or other anticoagulant supplements. Blood clogging can significantly reduce the capture efficiency as it will lower the interaction between rare cells and immobilized antibodies [33]. Second, the micropillars distribution will affect the encounter frequency between rare cells and micropillars. Optimized pillar size and pillar-pillar distance can maximize the interaction between rare cells and the micropillars [58]. Third, the capture of rare cells is largely determined by the surface marker (e.g., EpCAM) expression. Low marker expressed rare cells can escape from the immobilized antibodies. To counter this problem, multi-marker-based isolation can be used in this microfluidic system.

After the whole chip scan, the microfluidic device can be detached from the microscope slide, and the PDMS punch can be used to punch through the whole chip and dislodge the small PDMS piece containing the target cell. For this step, it is important to the punch straight to ensure the target cell is within punch zone. Following the target cell collection, the cell can either by cultured in cell culture medium for a short period of time or be lysed for proteomic analysis. As a sample preparation tool, the micropillar-based device is applicable for a variety of scMS technologies, such as label-free and multiplexed scMS [59,60], transferring identification based on filtering (TIFF) [61]. scMS enables the profiling of protein contents within an individual cell. Combined with NGS analysis, scMS can deepen our understanding of cellular states and dynamic and transcriptome changes within rare cell populations [62].

Overall, this work describe a microfluidic device that can be used for easily capture, detection and release of rare cells. The fabrication of the device is straightforward and reproducible. Due to the moderate shear rate inside the microfluidic device, the captured rare cells remain high viability. Therefore, the captured cell is amenable for culture and further analysis. Also, since the silicon molds are reusable, the micropillar-based device acquire the advantage of other PDMS-based devices—highly reproducible with relatively low cost per chip. PDMS materials will be the main consumables for replicating the micropillar-based devices. This microfluidic device will empower the field of single cell proteomics.

## Figures and Tables

**Figure 1 mps-06-00080-f001:**
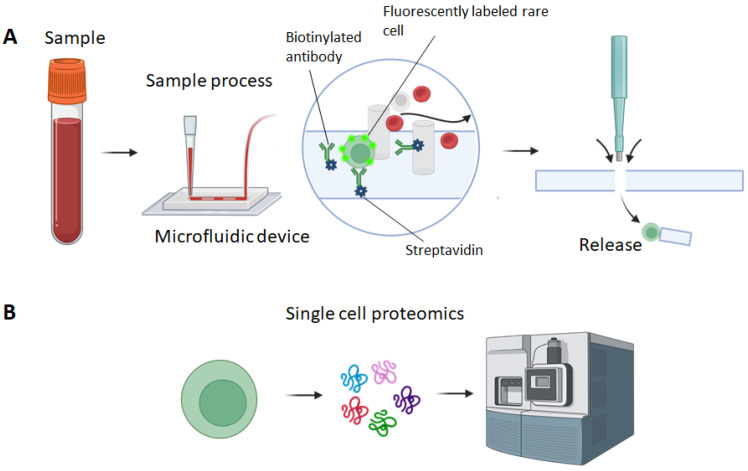
Illustration of rare cell isolation and release for single-cell proteomic analysis. (**A**) The microfluidic device containing a micropillar array is first immobilized with antibodies. The whole blood sample is then introduced to the micropillar-based device. Target rare cells are then specifically captured by antibodies in the microfluidic device. Following cell capture, the rare cells are detected and located. The rare cells are then released by punching through the microfluidic device. (**B**) The released rare cells are collected and used for single-cell proteomic analysis.

**Figure 2 mps-06-00080-f002:**
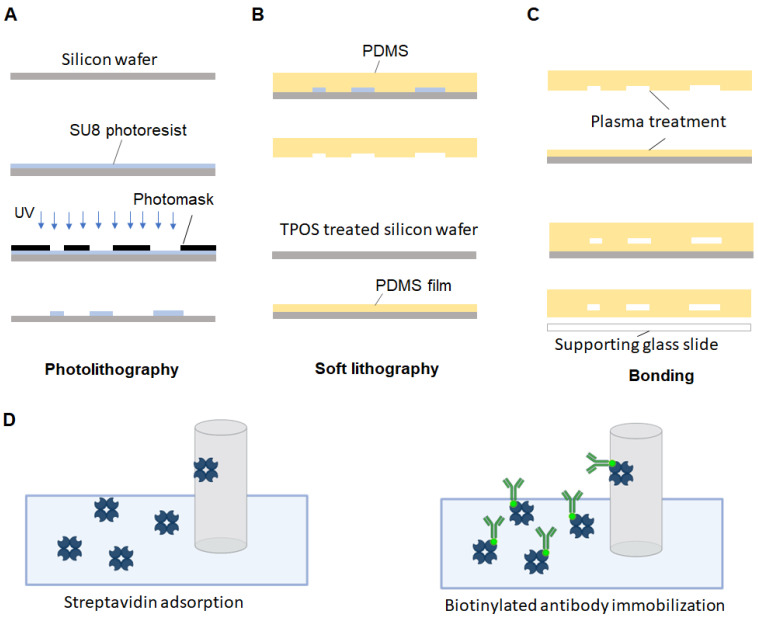
Illustration of micropillar-based device fabrication and functionalization. (**A**) The silicon mold is fabricated using photolithography. SU8 photoresist is spin coated on a silicon wafer. A chrome mask containing the micropillar array pattern is used to transfer the micropillar array feature to the SU8 layer. The silicon wafer is then developed to form silicon mold. (**B**) The PDMS substrate is formed by soft lithography using silicon mold. A thin PDMS film is made on a silicon wafer for device binding. (**C**) The PDMS substrate and the PDMS thin film are first plasma treated and then bonded together, the PDMS film bonded to the substrate is then detached from the silicon wafer. (**D**) The functionalization of the micropillar-based device includes two steps: physical adsorption of streptavidin and immobilization of biotinylated antibodies through streptavidin-biotin crosslinking.

## Data Availability

Not applicable.

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
