# Peer review of "A Micropillar Array Based Microfluidic Device for Rare Cell Detection and Single-Cell Proteomics"

_mps, 2023, doi:10.3390/mps6050080_

Round 1

Reviewer 1 Report

In the manuscript, Chen and Wang present a detailed protocol for fabricating an antibody-based microfluidic device that enables single-cell capture, release, and analysis. The authors introduce the experimental steps from four aspects: preparation of chromium mask, processing of silicon wafer mold, antibody modification of microfluidic chip, and capture of single cells in samples.

Overall, this protocol elaborates the fabrication of microfluidic chips for single-cell analysis, which is a great help for beginners. In my opinion, this article can be accepted for publication if the following concerns are well addressed.

1.     In the Introduction, the authors review some commonly used methods for single-cell microfluidic analysis. It is recommended that the authors add some of hydrodynamic methods for single-cell capture, such as 10.1002/anie.202302000

2.     In subsection 2.1, authors should give the model, brand, company, and country of the materials used. The information on material No. 15 is relatively comprehensive, while the information on other materials is somewhat lacking. The third material should be removed as it reappears later in the subsection 2.2 Equipment.

3.     In section 2.2, authors should use numbers instead of bullets. Moreover, on page 3 line 126, programmable, dual-channel infusion and withdrawal modular syringe pump is a new equipment and needs to be renumbered.

4.     When citing the experimental steps in the main text, please pay attention to use the correct step number. For example, page 5 line 193, “Step 1-4”, and page 6, line 262, “Step 18”. In addition, where is “Note 5” that stated by authors on page 4, line 169?

5.     The final antibody concentration should be given on page 5, line 234.

6.     By the way, the authors should proofread their manuscript carefully to avoid some grammatical and formatting errors throughout the paper, which will make the final manuscript more accessible.

Typos and grammatical mistakes

-      Page 1, line 16: “microfluidic-based” should be “microfluidics-based”.

-      Page 1, line 39: “proteomics analysis” should be “proteomic analysis”.

-      Page 2, line 49: “cane” should be “can”.

-      Note all superscripts and superscripts in the article, including but not limited to “N2”, “Ca2+”, “CO2”, “103 cells/mL”.

-      Please carefully check the capitalization of words in phrases or sentences. For instance, on page 3, lines 103-104, and on page 6, line 240.

-      Please check “e.g.”, and it should be in the unified format "e.g.,".

-      Note the correct writing of the units. Also, there must be a space between the number and the unit. For example, on page 4, line 180, “uL” should be “µL”, and on page 4, lines 170-177, spaces are missing between number and ℃.

By the way, the authors should proofread their manuscript carefully to avoid some grammatical and formatting errors throughout the paper, which will make the final manuscript more accessible.

Author Response

In the manuscript, Chen and Wang present a detailed protocol for fabricating an antibody-based microfluidic device that enables single-cell capture, release, and analysis. The authors introduce the experimental steps from four aspects: preparation of chromium mask, processing of silicon wafer mold, antibody modification of microfluidic chip, and capture of single cells in samples.

Overall, this protocol elaborates the fabrication of microfluidic chips for single-cell analysis, which is a great help for beginners. In my opinion, this article can be accepted for publication if the following concerns are well addressed.

  1. In the Introduction, the authors review some commonly used methods for single-cell microfluidic analysis. It is recommended that the authors add some of hydrodynamic methods for single-cell capture, such as 10.1002/anie.202302000

Response: Thank you for the suggestion. We added the referred paper as a reference (Ref. 26).

  1. In subsection 2.1, authors should give the model, brand, company, and country of the materials used. The information on material No. 15 is relatively comprehensive, while the information on other materials is somewhat lacking. The third material should be removed as it reappears later in the subsection 2.2 Equipment.

Response: We added all the information in subsection 2.1. The referred errors are corrected (highlighted). The referred material is removed.

  1. In section 2.2, authors should use numbers instead of bullets. Moreover, on page 3 line 126, programmable, dual-channel infusion and withdrawal modular syringe pump is a new equipment and needs to be renumbered.

Response: We changed the bullets to numbers.

  1. When citing the experimental steps in the main text, please pay attention to use the correct step number. For example, page 5 line 193, “Step 1-4”, and page 6, line 262, “Step 18”. In addition, where is “Note 5” that stated by authors on page 4, line 169?

Response: We corrected the errors. The Note 5 was marked in the early draft but was incorporated in the Expected Result.

  1. The final antibody concentration should be given on page 5, line 234.

Response: The concentration of antibodies is given.

  1. By the way, the authors should proofread their manuscript carefully to avoid some grammatical and formatting errors throughout the paper, which will make the final manuscript more accessible.

 Response: Thank you for the comment. We went through the manuscript and corrected the errors.

Typos and grammatical mistakes

-      Page 1, line 16: “microfluidic-based” should be “microfluidics-based”.

-      Page 1, line 39: “proteomics analysis” should be “proteomic analysis”.

-      Page 2, line 49: “cane” should be “can”.

-      Note all superscripts and superscripts in the article, including but not limited to “N2”, “Ca2+”, “CO2”, “103 cells/mL”.

-      Please carefully check the capitalization of words in phrases or sentences. For instance, on page 3, lines 103-104, and on page 6, line 240.

-      Please check “e.g.”, and it should be in the unified format "e.g.,".

-      Note the correct writing of the units. Also, there must be a space between the number and the unit. For example, on page 4, line 180, “uL” should be “µL”, and on page 4, lines 170-177, spaces are missing between number and â„ƒ.

Response: The typos and grammatical mistakes are corrected.

Reviewer 2 Report

The manuscript describes a protocol for a micropillar array-based microfluidic device for rare cell detection and single-cell proteomics. The study offers a highly efficient method for isolating, detecting, and collecting rare cells, such as circulating tumor cells, using a microfluidic-based approach. The method involves the use of a micropillar array-based microfluidic device for rare cell detection and single-cell proteomics. The device is designed to capture rare cells, such as circulating tumor cells, based on their unique surface markers using immobilized antibodies in the microchannels of the device. The micropillar array patterned in the microchannel increases the interaction between rare cells and antibodies, making them more likely to be captured. After sample processing, captured rare cells are detected within the microfluidic device using a fluorescence microscope. The identified rare cells are marked and released using a PDMS punch, and the collected rare cells are used for subsequent single-cell proteomic analysis. The method offers a highly efficient approach for isolating, detecting, and collecting rare cells, and has potential applications beyond rare cell analysis. I have the following comments/questions that authors may find useful.

What is the sensitivity and specificity of the method for detecting rare cells?

How reproducible is the method across different samples and users?

What is the limit of detection of the method for rare cells?

How does the method compare in terms of cost and time to existing methods?

Author Response

The manuscript describes a protocol for a micropillar array-based microfluidic device for rare cell detection and single-cell proteomics. The study offers a highly efficient method for isolating, detecting, and collecting rare cells, such as circulating tumor cells, using a microfluidic-based approach. The method involves the use of a micropillar array-based microfluidic device for rare cell detection and single-cell proteomics. The device is designed to capture rare cells, such as circulating tumor cells, based on their unique surface markers using immobilized antibodies in the microchannels of the device. The micropillar array patterned in the microchannel increases the interaction between rare cells and antibodies, making them more likely to be captured. After sample processing, captured rare cells are detected within the microfluidic device using a fluorescence microscope. The identified rare cells are marked and released using a PDMS punch, and the collected rare cells are used for subsequent single-cell proteomic analysis. The method offers a highly efficient approach for isolating, detecting, and collecting rare cells, and has potential applications beyond rare cell analysis. I have the following comments/questions that authors may find useful.

What is the sensitivity and specificity of the method for detecting rare cells?

Response: Since this is a proposed design and protocol, the actual sensitivity and specificity are not given. However, we gave a suggested sensitivity and specificity according to literature. We added a discussion in the Expected Result. (Page 8, 347-349)

How reproducible is the method across different samples and users?

Response: We added a discussion about factors that can affect the performance of the device in the Expected Result. (Page 8, 349-350)

What is the limit of detection of the method for rare cells?

Response: As mentioned in the Expected Result Section, the micropillar-based device can isolate less than 10 cells in 1 mL of blood.

How does the method compare in terms of cost and time to existing methods?

Response: We added a short discussion in the Expected Result (Page 9, Line 376-379).

Reviewer 3 Report

In the manuscript,” A Micropillar Array Based Microfluidic Device for Rare cell Detection and Single-Cell Proteomics”, the authors presented a microfluidic method having micropillars for analysing rare cells, such as circulating tumor cells (CTCs) and rare immune cells for single-cell proteomic studies. The method looks reasonably detailed, and the results should be promising which can be used for identifying and capturing rare cells. This manuscript is appropriate to be published after the following minor revisions. The comments for this manuscript are given below:

1. Will the blood injected into the microfluidic device also require anti-coagulant? If so, it should be mentioned.

2. It has been mentioned that the capture efficiency should be more than 90%? Can references be provided for micropillars-based capture devices which used a similar approach? What are the potential factors that might lower the capture efficiency? For instance, do all CTCs express EpCAM?

3. A more detailed diagram of the schematic representation of the capture of CTCs expressing EpCAM on the microchannel’s surface should be provided in Fig. 1A. The primary and secondary antibodies and the fluorophores should be depicted. Also, an illustration of the functionalization of the PDMS surfaces should be provided.

4. Which instruments are used for proteomic analysis? What are the applications of proteomic analysis and why is it important?

5. Why is Trichloro(1H,1H,2H,2H-perfluorooctyl)silane (TPOS) used for surface modification of PDMS? Was it also used for creating hydrophobic surfaces withing PDMS similar to HMDS?

Moderate editing of English language required.

Author Response

Reviewer 3

In the manuscript,” A Micropillar Array Based Microfluidic Device for Rare cell Detection and Single-Cell Proteomics”, the authors presented a microfluidic method having micropillars for analysing rare cells, such as circulating tumor cells (CTCs) and rare immune cells for single-cell proteomic studies. The method looks reasonably detailed, and the results should be promising which can be used for identifying and capturing rare cells. This manuscript is appropriate to be published after the following minor revisions. The comments for this manuscript are given below:

  1. Will the blood injected into the microfluidic device also require anti-coagulant? If so, it should be mentioned.

Response: We added information about ant-coagulant. (Page 6, Line 265)

  1. It has been mentioned that the capture efficiency should be more than 90%? Can references be provided for micropillars-based capture devices which used a similar approach? What are the potential factors that might lower the capture efficiency? For instance, do all CTCs express EpCAM?

Response: We added references to support this claim about capture efficiency. We added a discussion in the Expected Result. (Page 347-360)

  1. A more detailed diagram of the schematic representation of the capture of CTCs expressing EpCAM on the microchannel’s surface should be provided in Fig. 1A. The primary and secondary antibodies and the fluorophores should be depicted. Also, an illustration of the functionalization of the PDMS surfaces should be provided.

Response: We edited Fig. 1A and added the functionalization of the PDMS surfaces on Fig. 2D.

  1. Which instruments are used for proteomic analysis? What are the applications of proteomic analysis and why is it important?

Response: We added a short discussion in the Expected Result Section about proteomic analysis. (Page 9, 366-371)

  1. Why is Trichloro(1H,1H,2H,2H-perfluorooctyl)silane (TPOS) used for surface modification of PDMS? Was it also used for creating hydrophobic surfaces withing PDMS similar to HMDS?

Response: The TPOS is used for the treatment of silicon mold. It functions similar to HMDS. It is used to ensure a hydrophobic surface on the silicon mold. It will prevent cured PDMS stick to the silicon mold surface.